# Comparison of Photocatalytic Biocidal Activity of TiO₂, ZnO and Au/ZnO on *Escherichia coli* and on *Aspergillus niger* under Light Intensity Close to Real-Life Conditions

Mohamad Al Hallak [1], Thomas Verdier [1], Alexandra Bertron [1], Kevin Castelló Lux [1], Ons El Atti [2], Katia Fajerwerg [2], Pierre Fau [3], Julie Hot [1], Christine Roques [4] and Jean-Denis Bailly [5,6,*]

1 Laboratoire Matériaux et Durabilité des Constructions (LMDC), INSA Toulouse, 135 Avenue de Rangueil, 31400 Toulouse, France; alhallak@insa-toulouse.fr (M.A.H.); tverdier@insa-toulouse.fr (T.V.); bertron@insa-toulouse.fr (A.B.); k.castello-lux@hotmail.com (K.C.L.); hot@insa-toulouse.fr (J.H.)
2 Laboratoire de Chimie de Coordination (LCC), Université de Toulouse, CNRS 205 Route de Narbonne, 31400 Toulouse, France; ons.elatti@lcc-toulouse.fr (O.E.A.); katia.fajerwerg@lcc-toulouse.fr (K.F.)
3 Laboratoire de Physique et Chimie de Nano-Objets (LPCNO), INSA Toulouse, 135 Avenue de Rangueil, 31400 Toulouse, France; pfau@insa-toulouse.fr
4 Laboratoire Génie Chimique (LGC), Université de Toulouse, CNRS, INPT, UPS, 35 Chemin des Maraîchers, 31400 Toulouse, France; roques730@aol.com
5 École Nationale Vétérinaire de Toulouse, 23 Chemin des Capelles, CEDEX, 31076 Toulouse, France
6 Laboratoire de Chimie Agro-Industrielle (LCA), Université de Toulouse, INRA, INPT, 4 Allées Emile Monso, 31030 Toulouse, France
* Correspondence: jean-denis.bailly@envt.fr

**Abstract:** Microbial contamination of the surface of building materials and subsequent release of microbial particles into the air can significantly affect indoor air quality. Avoiding the development or, at least, reducing the quantity of microorganisms growing on building materials is a key point to reduce health risks for building occupiers. In that context, the antimicrobial activity of TiO₂, ZnO and Au/ZnO was assessed by measuring log reductions of *Escherichia coli* and *Aspergillus niger* populations both in the dark and under a light intensity close to real-life conditions. The bactericidal activities (≥2.3 log reduction) of tested products were stronger than their fungicidal activities (≤1.4 log reduction) after 2 h of contact. Different parameters including concentration of photocatalyst, intensity of light (dark vs. 5 W/m² UV-A), and duration of contact between photocatalyst and microbial cells and spores were investigated. Results of this study confirmed bactericidal activities of TiO₂, ZnO and AuZnO on *E. coli* and brought new insight on their fungicidal activity on the spores of *A. niger*. They also confirmed the greatest antimicrobial efficiency of ZnO compared to TiO₂ and its increased photocatalytic activity when decorated with Au, leading to the highest log reductions detected after 2 h of contact for both tested microorganisms (4 and 1.4 for *E. coli* and *A. niger*, respectively). The antimicrobial activity was enhanced by the duration of contact between microorganisms and nanoparticles of the different tested photocatalytic products.

**Keywords:** bactericidal activity; fungicidal activity; indoor air; photocatalysis; TiO₂; ZnO; AuZnO

## 1. Introduction

Indoor air pollution is an important cause of diverse health problems for occupants, including respiratory diseases, allergic symptoms, cancers and cardiovascular problems [1–7]. For several years now, there is an increased awareness to the effect of indoor air quality on human health and wellbeing [7,8]. Actually, people spend 80–90% of their times indoors [9,10] emphasizing the importance of understanding the main causes of indoor air pollution and finding suitable solutions to improve indoor air quality. In 2009, The World Health Organization (WHO) reported that biological pollution is one of the main causes of the degradation of indoor quality [11]. Investigations on microbial contamination on the

surfaces of building materials have highlighted that it may also have a significant impact on the microbial communities present in indoor air [7,12]. However, the quantitative assessment of this impact has been little studied. When favorable conditions (mainly humidity and nutrient content) are present, building materials can allow growth of microorganisms initially present or brought in by activities and environmental conditions [7,12]. Upon their development on surfaces, microorganisms produce particles such as spores, toxins or volatile organic compounds and other metabolites that can be aerosolized and inhaled by occupants [13–16]. To prevent or to reduce microbial contamination on surfaces, the antimicrobial activity of several chemical products has already been studied, i.e., semiconductors such as titanium dioxide ($TiO_2$), zinc oxide (ZnO), gold zinc oxide (AuZnO), gallium arsenide, tungsten (VI) oxide ($WO_3$), gallium phosphide, and cadmium as well as fatty acids and glycerol esters [17–19]. Photocatalysis approach allows the transfer of solar energy, received by a semiconductor, into chemical energy [20]. The process occurs when a photocatalyst is excited by light with an energy higher than its band gap, inducing the formation of energy-rich electron-hole pairs that can be involved in redox reactions with a lethal effect at the cell level [21,22].

Titanium dioxide ($TiO_2$) is widely recognized as one of the most efficient photocatalysts used for air purification [17,18]. It has adequate optical and electronic properties, high-photocatalytic activity, and high-chemical stability. It is commonly used as reference material for photocatalysis applications [23]. In addition to being non-toxic, this compound is inexpensive, available, and abundant [18]. Under "high" intensities of light, over >10 W/m$^2$, the photocatalysis of $TiO_2$ nanoparticles induced antimicrobial activity against a wide variety of microorganisms, including algae, viruses, bacteria and fungi [24–27]. Nevertheless, few studies have been carried out in the last five years on the effect of $TiO_2$ on microorganisms at lower intensity levels ($\leq 5$ W/m$^2$), closed to real-world conditions (30 W/m$^2$ on sunny days and 5–10 W/m$^2$ on cloudy days outdoors and 4–5 W/m$^2$ indoors) [17,28–30].

Zinc oxide (ZnO) is also a well-known photocatalyst that exhibits high photosensitivity, excellent stability, low toxicity, and a bandgap similar to that of $TiO_2$ (approximately 3.3 eV for ZnO wurtzite and 3.2 eV for $TiO_2$ anatase). It is also cost-effective and has a higher oxidation capacity. ZnO is a semiconductor metal oxide that is of increasing interest due to its various morphologies such as nanorods, nanotubes, and nanowires, each with unique properties depending on the synthesis method [31,32]. The photocatalytic activity of ZnO can be extended to the visible light region by the addition of metallic nanoparticles (NPs) such as silver (Ag), copper (Cu) and gold (Au). The introduction of these metals induces a localized surface plasmon resonance (LSPR) effect and enhances charge carrier separation, thereby reducing recombination. In the literature, most applications of ZnO-based systems focus on the photodegradation of dyes or organic pollutants such as rhodamine, methylene blue, and Congo red [33,34]. At higher light intensities, the photocatalytic biocidal activity of ZnO [35,36] and metal supported ZnO nanocomposites, such as CuO/ZnO [37,38], Ag/ZnO [39,40] and Au/ZnO [41,42], were assessed over different bacteria, such as *Escherichia coli, Staphylococcus aureus, Bacillus atrophaeus*, and fungi such as *Aspergillus niger* and *Candida albicans*. The bactericidal performances were generally attributed to the combined effects of mechanical piercing of ZnO nanoparticles [36] (nanorods, spheroide, nanoflowers, etc.), ions release ($Zn^{2+}$, $Cu^{2+}$, $Cu^+$, $Ag^+$) [37,38], and effective generation of reactive oxygen species due to a higher charge separation [40–43].

The present study is a preliminary work within the framework of developing photocatalytic products to prevent microbial contamination and growth of microorganisms on surfaces of indoor building materials. It aims to compare the efficiency of pure photocatalysts ($TiO_2$, ZnO, Au/ZnO), i.e., without interfering material (such as organic binder, tensioactive...) in a neutral activation media: water. The experimental design was firstly designed to compare the bactericidal and fungicidal activities of the three selected compounds by direct contact on two frequent indoor contaminants (*E. coli* cells and *A. niger* spores), using a suspension test, as a previously described approach at European level

(EN 14885), for testing antiseptics and disinfectants [43]. In order to be close to indoor activation conditions, chosen light intensity was relatively low (<10 W/m$^2$) compared to light intensities used in previously available studies. The impact of the concentration of the photocatalyst as well as the duration of exposure were evaluated. The development of fungicidal and/or bactericidal photocatalytic compounds could be of great interest in preventing and reducing the microbial contamination of building materials and subsequently improving indoor air quality.

## 2. Results

### 2.1. Characterization of the Photocatalysts

Figure 1 shows the X-Ray diffraction analysis (XRD) patterns of TiO$_2$, ZnO and Au/ZnO metal oxide semiconductors. The peaks were indexed by using the Inorganic Crystal Structure Database (ICSD). Anatase (Joint Committee on Powder Diffraction Standards JCPDS 021-1272) crystalline phase was detected for TiO$_2$ powder. Phase quantification Rietveld refinement of this X-ray diffractogram using the High X'pert software (Malvern Panalytical, https://www.malvernpanalytical.com accessed on 19 July 2023) provided 83% anatase and 17% rutile, consistent with literature values for P25 TiO$_2$ [44]. Concerning ZnO, the X-ray pattern was consistent with that of the hexagonal wurtzite crystalline phase (JCPD 036-1451), the most stable in ambient conditions [45]. The average crystallite sizes were estimated from the Scherrer formula to be equal to 30, 40 and 60 nm for ZnO, TiO$_2$ anatase and rutile, respectively. The XRD pattern of Au/ZnO showed characteristic peaks of Au in addition to those of ZnO. As expected, the diffraction intensity of Au peaks (38.2° and 44.2°) was low due to the small size and low weight content of Au nanoparticles (NPs). The weight content of Au measured by EPMA was equal to 1.25 $\pm$ 0.19 wt% close to the 1 wt% nominal amount, which confirmed the good efficiency of the photo-deposition technique to master the Au content in the composite material.

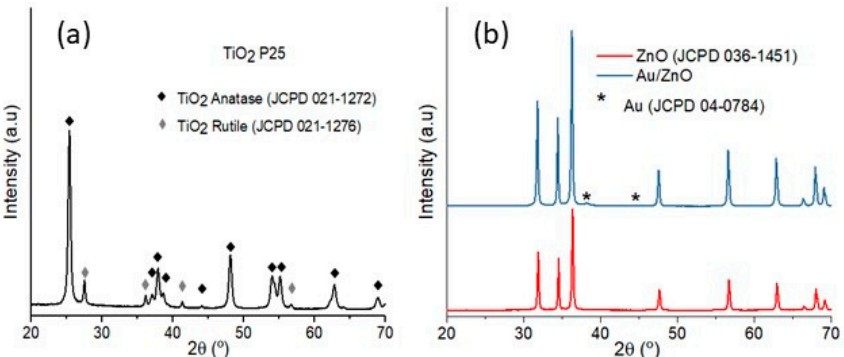

**Figure 1.** X-ray diffractograms of (**a**) TiO$_2$ P25 and (**b**) ZnO and Au/ZnO. Peaks were indexed by using the Inorganic Crystal Structure Database (ICSD).

The (Brunauer–Emmett–Teller) BET specific surface area's obtained were: ca 50 m$^2$/g for TiO$_2$ and ca 14 m$^2$/g for ZnO and Au/ZnO. The shape of the photo-deposited Au NPs was further investigated through Transmission Electron Microscopy (TEM) and High-Resolution Transmission Electron Microscopy (HRTEM) analyses (Figure 2). The size was accessed by measuring the longest dimension of Au NPs. Polymorphous Au NPs (nanorods, triangular and spherical NPs) were observed with a mean diameter of 30 $\pm$ 13 nm. The HRTEM images (Figure 2b,c) show a well-defined interfacial contact between Au NP and ZnO, which is known to play a crucial role in photocatalytic reactivity.

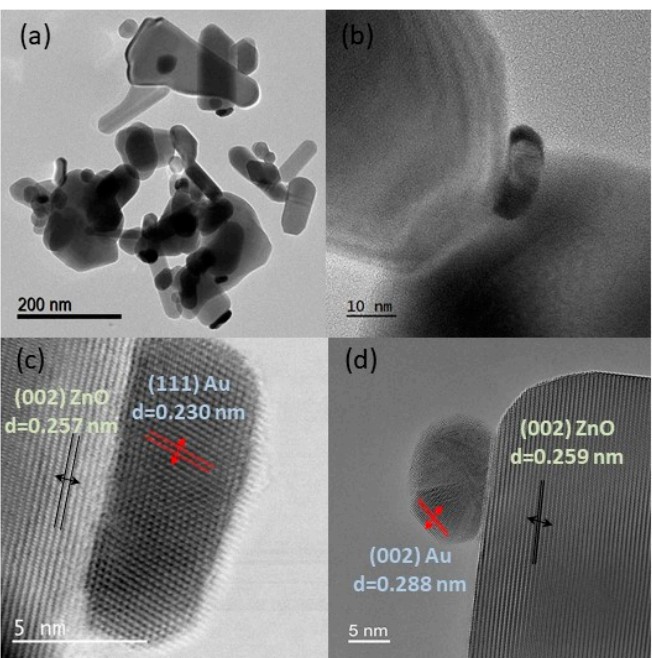

**Figure 2.** HRTEM images of Au/ZnO highlighting various morphologies of Au NPs (**a**,**b**); HRTEM images of Au/ZnO interface (**c**,**d**).

### 2.2. Bactericidal Effects of Photocatalysts on Escherichia coli

#### 2.2.1. Influence of Light

To compare the bactericidal activities of $TiO_2$, ZnO and Au/ZnO against *E. coli* CIP 53126, experiments were first carried out at a concentration of 10 g/L in the dark and under a light intensity of 5 $W/m^2$. For all photocatalysts and in both experimental conditions, the bactericidal effect increased with time and the maximal effect was observed after 2 h of incubation (Figure 3). In the dark, the average $\log_{10}$ reductions of viable *E. coli* cells after 2 h of contact were 2.27 ± 0.08, 3.36 ± 0.01 and 3.04 ± 0.01 for $TiO_2$, ZnO and Au/ZnO, respectively (Figure 3a). Under light, the average $\log_{10}$ reductions were 2.58 ± 0.08, 3.43 ± 0.02 and 3.98 ± 0.02 for $TiO_2$, ZnO and Au/ZnO, respectively (Figure 3).

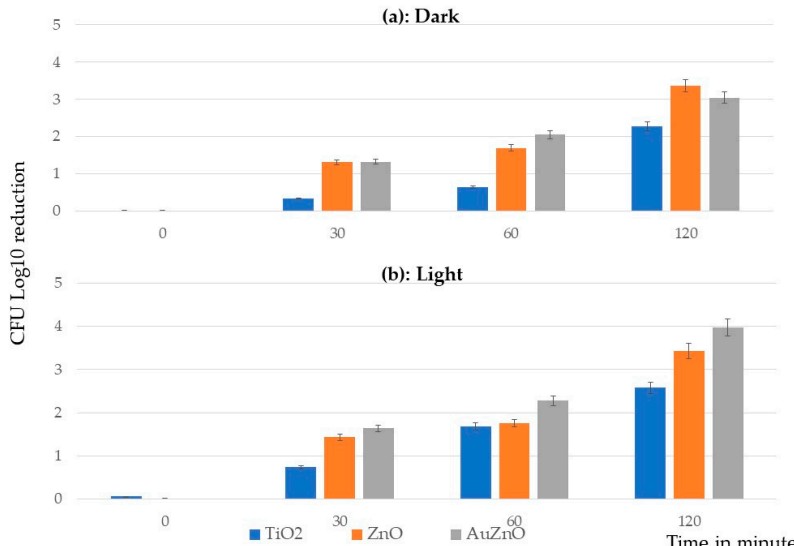

**Figure 3.** Average log reduction of *E. coli* 53126 after contact with different photocatalysts $TiO_2$ (blue bars), ZnO (orange bars) and AuZnO (grey bars) at 10 g/L concentration in the dark (**a**) or under an exposition to a light intensity of 5 $W/m^2$ (**b**). Results are expressed as mean ± SD (*n* = 2).

In the dark, ZnO and Au/ZnO both consistently showed higher levels of reduction compared to TiO$_2$, whatever the time point tested. Additionally, Au/ZnO exhibited the highest average reduction in log concentration at t = 60 min, while ZnO showed the highest reduction at t = 120 min.

In the light, a slight increase in bactericidal activity was observed for the three photocatalysts from 30 min of contact. The highest gain in activity was observed for Au/ZnO (about 1 log).

### 2.2.2. Influence of Photocatalyst Concentration

To investigate the influence of the concentration of photocatalysts on their bactericidal activity against *E. coli*, another set of experiments was carried at 1 g/L. For these assays, incubation was prolonged to 4 h. As previously observed at 10 g/L, the bactericidal effect increased with time of incubation for the three photocatalysts tested.

In the dark, the bactericidal activity was strongly weaker in comparison to assays performed with a concentration of 10 g/L. Indeed, after 4 h of incubation, the maximum reductions were only 27 ± 4%, 48 ± 1% and 37 ± 1% of those observed at 10 g/L for TiO$_2$, ZnO and Au/ZnO, respectively (Figure 4a,c,e). By contrast, under light, the prolongation of incubation until 4 h allowed to reach similar bactericidal activity as those observed after 2 h of contact at 10 g/L for the three photocatalysts (Figure 4b,d,f). Indeed, in that condition, the reductions of *E. coli* population were 97 ± 2%, 99 ± 1% and 98 ± 1% of those observed at 10 g/L for TiO$_2$, ZnO and AuZnO, respectively.

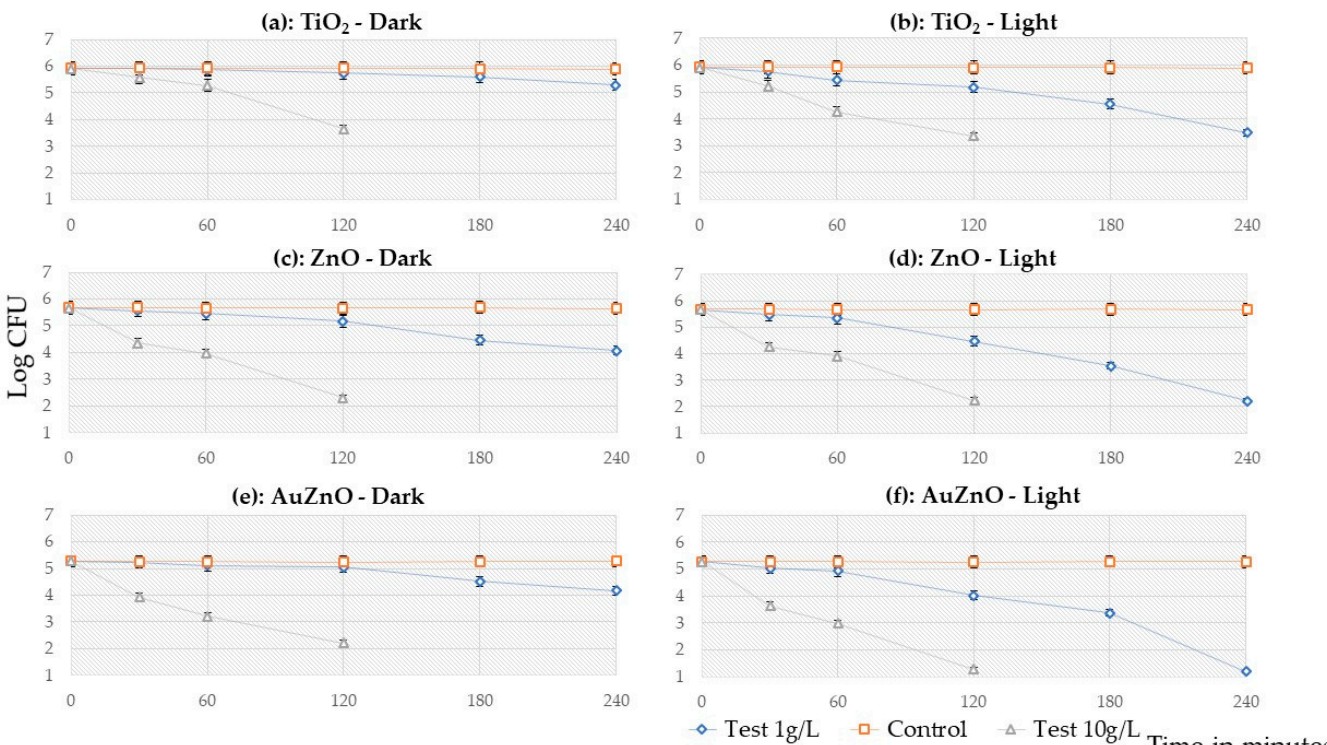

**Figure 4.** Comparison of time-dependent bactericidal activities (*E. coli* 53126) of photocatalysts used at 1 and 10 g/L. Orange curves: control tubes; grey curves: 10 g/L of photocatalyst; blue curves: 1 g/L of photocatalyst. TiO$_2$ in the dark (**a**), TiO$_2$ under light (**b**), ZnO in the dark (**c**), ZnO under light (**d**), Au/ZnO in the dark (**e**) and Au/ZnO under light (**f**). The results presented are expressed as mean ± SD of two independent experiments.

### 2.3. Fungicidal Effects of Photocatalysts on Aspergillus niger

2.3.1. Influence of Light

As conducted for the bactericidal experiments, the fungicidal activities of TiO$_2$, ZnO and Au/ZnO on *A. niger* spores were firstly determined at a concentration 10 g/L in the dark and under a light intensity of 5 W/m$^2$.

The three photocatalysts, TiO$_2$, ZnO and Au/ZnO, provided weak fungicidal activity in the dark. Even after 120 min of contact, log reductions were not significant ($\leq$0.11) (Figure 5a).

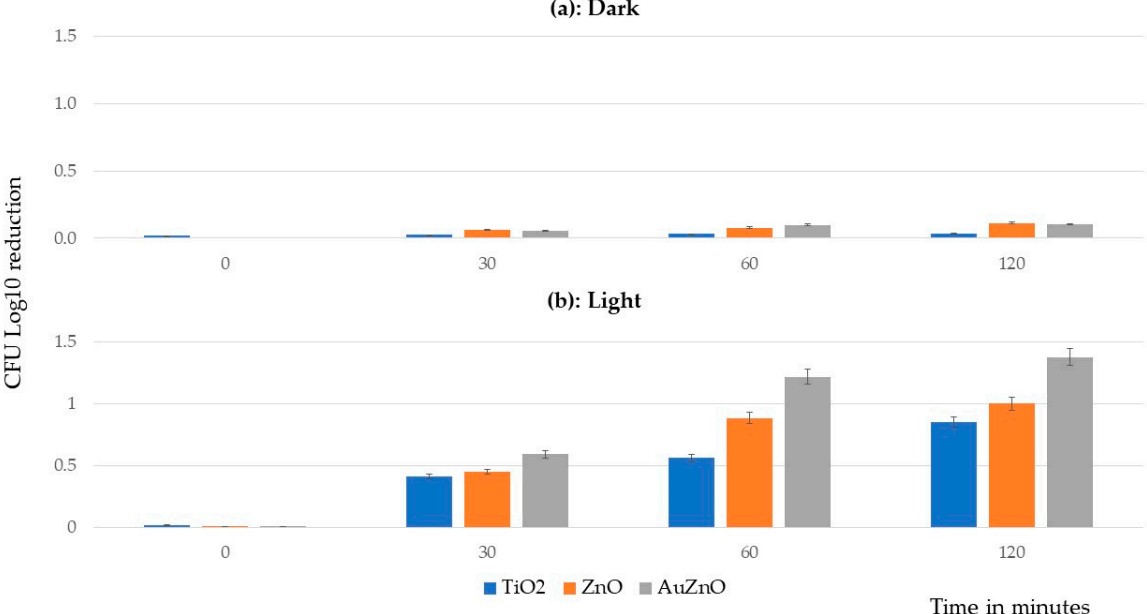

**Figure 5.** Time-dependent average log reduction of *A. niger* CBS 733.88 after contact TiO$_2$ (blue bars), ZnO (orange bars) and AuZnO (grey bars) at 10 g/L concentration in the dark (**a**) or under an exposition to a light intensity of 5 W/m$^2$ (**b**). Results are expressed as mean $\pm$ SD (*n* = 2).

Under light, the average log$_{10}$ reductions were mildly increased (Figure 5b). After 30 min of contact, the fungicidal activities of photocatalysts significantly increased, with average reductions of 0.41 $\pm$ 0.01, 0.45 $\pm$ 0.01 and 0.59 $\pm$ 0.01 for TiO$_2$, ZnO and Au/ZnO, respectively. After 60 min and until 120 min of contact, the fungicidal activities of all photocatalysts continued to increase, showing an average reduction of contact of 0.85 $\pm$ 0.03, 1.00 $\pm$ 0.01 and 1.38 $\pm$ 0.03 for TiO$_2$, ZnO and Au/ZnO, respectively. At each time point, ZnO exhibited a higher average reduction in log concentration than TiO$_2$, while Au/ZnO showed the highest reduction among the three photocatalysts.

The reduction in fungal spore concentration was therefore higher under light than in the dark for all three photocatalysts. As for bactericidal activity, the presence of light enhanced the fungicidal activity of tested photocatalysts with bactericidal activities being clearly higher than fungicidal ones.

2.3.2. Influence of Photocatalyst Concentration

The influence of the combination concentration of the photocatalysts' contact time on their fungicidal activity against *A. niger* was also tested through another set of experiments carried at 1 g/L for ZnO and Au/ZnO with exposition up to 4 h (Figure 6).

As expected, the fungicidal activities of ZnO and Au/ZnO at 1 g/L were negligible in the dark after 4 h of contact. Under light, the fungicidal activities of ZnO and Au/ZnO at 1 g/L were enhanced but, even after 4 h of contact, did not reach those obtained after 2 h of contact at 10 g/L concentration, as was the case with the *E. coli* population. The reduction

of *A. niger* spores were 64 ± 2% and 78 ± 3% of those observed at 10 g/L for ZnO and Au/ZnO, respectively.

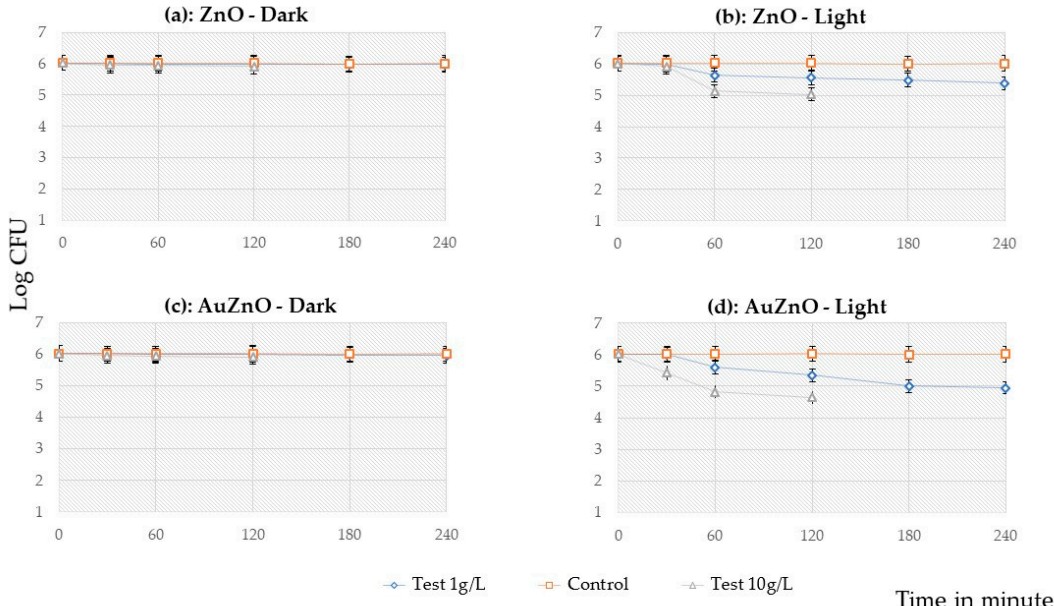

**Figure 6.** Comparison of time-dependent fungicidal activities (*A. niger* CBS 733.88) of photocatalysts used at 1 and 10 g/L. Orange curves: control tubes; grey curves: 10 g/L of photocatalyst; blue curves: 1 g/L of photocatalyst. ZnO in the dark (**a**), ZnO under light (**b**), Au/ZnO in the dark (**c**) and Au/ZnO under light (**d**). The results presented are the mean ± SD of two independent experiments.

### 2.3.3. Additional Test on *A. niger* with $TiO_2$

$TiO_2$ at 10 g/L had the weakest fungicidal activity compared to ZnO and Au/ZnO. To investigate this lack of activity in the tested conditions, another series of experiments was carried out with an incubation prolonged to 24 h of contact between *A. niger* spores and 10 g/L $TiO_2$ under light. Figure 7 shows that the limited reduction observed after 2 h was not increased after 24 h, confirming the resistance of *A. niger* to the photocatalytic effect of $TiO_2$.

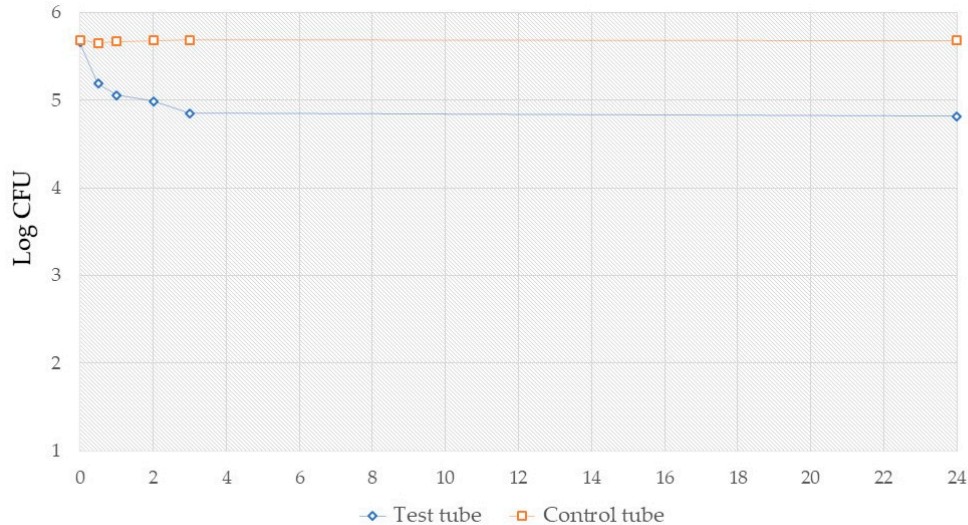

**Figure 7.** Effect of 10 g/L of $TiO_2$ on spores of *A. niger* throughout a 24 h incubation period under light. Orange curve: control tube; Blue curve: test tube. One representative experiment is shown.

## 3. Discussion

### 3.1. Effects of Photocatalyts on E. coli Cells

The effect of photocatalysts on *E. coli* was evaluated both in the dark and under light conditions. In the dark, ZnO showed the highest bactericidal activity, followed by Au/ZnO and TiO$_2$. No studies have reported the bactericidal activity of Au/ZnO in the dark. The decoration of ZnO by Au NPs is known to enhance its photocatalytic activity [46] as the implanting of other metals such as cobalt to form Co doped ZnO and other metals at different suitable concentrations [47]. These findings are consistent with previous studies carried out with ZnO [48,49] and TiO$_2$ [50,51]. The bactericidal activity of ZnO (*E. coli* population tested) was notably reported to be stronger than that of TiO$_2$ by Jones et al. [52]. In addition, ZnO was reported to own significant bactericidal effects on a wide range of microorganisms [48,49]. Indeed, the bactericidal activities observed may be explained by the ability of well-dispersed nanoparticles of photocatalysts used to interact with bacterial cells, absorbing them to their surfaces with a concentration and time-dependent effect [53].

Under light, the log reduction obtained with Au/ZnO was the strongest (R = 3.98 $\pm$ 0.01 and 3.91 $\pm$ 0.01 for 10 g/L and 1 g/L, respectively) followed by ZnO (R = 3.43 $\pm$ 0.02 and 3.44 $\pm$ 0.02 for 10 g/L and 1 g/L, respectively) and then TiO$_2$ (R = 2.58 $\pm$ 0.08 for 10 g/L). These results agree with the fact that the bactericidal activity of photoactivated ZnO is enhanced when decorated with Au NPs [46]. Moreover, ESR measurements carried out by Kevin Castelló Lux et al. showed a generation of radical oxygen species for both ZnO and Au/ZnO), higher for Au/ZnO [28,54].

These results are in accordance with a previous study reporting strong bactericidal activity of ZnO on *E. coli* and on *Listeria monocytogenes* where authors obtained seven log reduction after only one hour of incubation at a light intensity up to 96 W/m$^2$ [55]. Moreover, a recent study investigated the antimicrobial effect of Au doped ZnO decorated on single wall nanotubes and multi wall nanotubes of carbon against *E. coli* and *Staphylococcus aureus* at a light intensity of 8 W/m$^2$, which was close to the intensity used in this study (5 W/m$^2$) [56]. The log reduction values, R, obtained in this study were between 4 and 4.5 in log10 after 24 h of incubation, similar to the values reported here (3.98 for Au/ZnO after 2 h of incubation). Indeed, at a lower concentration (1 g/L), four hours of contact with *E. coli* cell suspension were required to observe bactericidal activities combined with photocatalytic activities that were approximately similar to those obtained at 10 g/L after two hours of contact, confirming also a concentration and time-dependent bactericidal activity. As an example, a concentration of TiO$_2$ as low as 0.1 g/L showed a strong bactericidal activity but after 24 h of contact using a light intensity range of 14–55 W/m$^2$ [57]. The ability of the photocatalytic products used to damage the cell membrane of *E. coli* may explain their high-bactericidal activity [18,58].

### 3.2. Effects of Photocatalysts on A. niger Spores

In contrast to their bactericidal effects on *E. coli*, the photocatalysts tested were found to have a negligible fungicidal activity against *A. niger* in the dark and to be less effective under light.

In the dark, even at a higher concentration, the tested photocatalysts demonstrated no fungicidal activities against *A. niger*. This observation implied that there was no efficient interaction between photocatalyst nanoparticles and fungal spores of *A. niger* [50,59]. The resistance of *A. niger* against TiO$_2$ observed after 24 h of contact in the dark was previously reported by KP Yu et al. [50]. However, several studies on the fungicidal activity of ZnO and decorated ZnO in the dark, reported the ability of ZnO [60] and nickel-doped ZnO (NiZnO) to inhibit the growth of yeasts at higher concentrations [61]. The authors suggested that a minimal fungicidal concentration of 1 g/L ZnO together with 5 mM of Histidine amino acids was responsible for growth inhibition of *Candida albicans* [60].

Under light, experiments carried out at a higher concentration (10 g/L) provided low fungicidal activity of Au/ZnO, ZnO and TiO$_2$ photocatalysts (R = 1.38 $\pm$ 0.03 for Au/ZnO R = 1.00 $\pm$ 0.01 for ZnO where R= 0.85 $\pm$ 0.03 for TiO$_2$). Notably, previous studies have

reported higher fungicidal activity of $TiO_2$ mixed with Ag NPs against *A. niger*, but at higher light intensities of 14–55 $W/m^2$ [57]. These results agree with the fact that the fungicidal activity of photoactivated ZnO is enhanced when decorated with Au NPs [46].

The lack of activity of $TiO_2$ on *A. niger* within light intensities, close to real-life conditions, has not been reported in the literature yet. One investigation has shown the ability of $TiO_2$ to inhibit the growth of *A. niger* on woods, but these results were obtained at very high light intensities [59]. It is important to highlight that previous findings showed higher fungicidal activity of $TiO_2$ mixed with Ag NPs on *A. niger* but these results were again obtained using a very high light intensity of 40 $W/m^2$ [62] or by continuous UV irradiation for 20 days [59]. For ZnO and metal-doped ZnO, previous papers reported their effectiveness on yeasts growing on plants and the possibility to use them in agriculture for plant protections from pathogenic yeasts [40,42,60,63]. A previous study showed the strong fungicidal activity of ZnO activity against *A. niger* [45] and *Botrytis cinerea*, a fungus that affects many plant species, but at a very high light intensity of 96 $W/m^2$ [55]. Very few studies presented the fungicidal activity of metal-doped/decorated ZnO. A study reported strong fungicidal activity of Cu-doped ZnO against yeasts but again at higher light intensities >22 $W/m^2$ [40]. In previous studies, high-fungicidal activities of photocatalysts were indeed achieved, but predominantly on yeasts. The observed differences in the effects of $TiO_2$, ZnO, and Au/ZnO on fungal and bacterial cells can be attributed to the variations in their cell membrane compositions [64,65].

### 3.3. Implications for Practical Uses of Photocatalysts

The results obtained in this study suggest that photocatalysts could potentially be employed for controlling *E. coli* in appropriate conditions. The effectiveness of photocatalysts under realistic light conditions was evaluated, considering the differential effects on bacterial cells and fungal spores. While the photocatalysts showed promising results in controlling *E. coli* bacteria, their effectiveness against *A. niger* was limited.

Our findings raise important implications for practical applications. For instance, in antimicrobial treatments, photocatalysts can be utilized for bacterial control, but alternative methods might be required to target fungal spores effectively. Further research is required to optimize the application conditions and explore approaches to enhance the destruction of fungal cells and prevent secondary growth. Additionally, investigations can focus on the development of novel photocatalysts. Developing novel photocatalysts involves designing materials with specific properties that can optimize their photocatalytic activity and antimicrobial efficacy. Researchers can explore various approaches, such as modifying the composition, structure, or surface properties of existing photocatalysts, or even developing entirely new materials. For example, researchers may focus on optimizing the bandgap energy of the photocatalyst, which determines the wavelengths of light that can be effectively utilized for photocatalytic reactions. Fine-tuning the bandgap energy can enhance the absorption of light and increase the generation of reactive oxygen species, ultimately improving the antimicrobial activity. Moreover, researchers may investigate the incorporation of different metal or non-metal elements into photocatalytic materials to enhance their photocatalytic properties. Doping or alloying can modify the electronic structure, band alignment, or charge transfer characteristics, resulting in improved photocatalytic performance and antimicrobial effects.

Finally, if tests performed using aqueous solutions of microorganisms are useful to screen and compare the activity of different photocatalysts toward microorganisms of interest without any interference; future experiments shall focus on the evaluation of their biocidal activity at similar light intensities on solid surfaces.

## 4. Materials and Methods

### 4.1. Microbial Strains

Two of the most frequent microorganisms identified on building materials were chosen to evaluate the antimicrobial activities of $TiO_2$, ZnO and Au/ZnO nanoparticles: the mould *A. niger* [7] and the Gram negative bacteria *E. coli* [66].

*E. coli* CIP 53126 was obtained from the collection of the Pasteur Institute (CIP, Paris, France) and the *A. niger* CBS 733.88 strain was obtained from the Westerdijk Fungal Biodiversity Institute (CBS, Westerdijk Fungal Biodiversity Institute, Utrecht, The Netherlands). Strains were stored at $-80\ ^\circ$C in Eugon medium (Biomérieux, Craponne, France) supplemented with 10% glycerol.

Bacterial cells were pre-cultured for 16–24 h at 36 $^\circ$C on trypticase soy agar (TSA, Biomérieux, Craponne, France) before each experiment. For tests, bacterial cells were dispersed in 1/500 nutrient broth medium and the bacterial suspension was adjusted to about 108 cells/mL by spectrophotometry ($\lambda$= 640 nm). The cell suspension concentration was checked by determining the number of CFU/mL after dilution and inclusion in TSA.

For fungal suspension, *A. niger* strain was cultured on Sabouraud agar medium (SAB, Biomérieux, Craponne, France) at 22.5 $^\circ$C for 10 to 14 days to prepare the spore suspension [67]. After incubation, 10 mL of sterilized distilled water (90%) + Tween 80 (10%) with sterile glass beads were added into the flask and gently stirred for at least 2 min. Then, the liquid phase was collected and filtered through sterile frit 080557-2 (40–100 μm) into a sterile pot containing sterile glass beads to prevent spores from clustering. The concentration of the spore suspension was determined by counting spores on Malassez cell via optical microscope and adjusted to about $10^7$ cells/mL. The cell suspension concentration was checked by determining the number of CFU/mL after dilution and inclusion in Sabouraud agar.

### 4.2. Photocatalytic Materials

#### 4.2.1. Commercial Oxides

Two bare commercial metal oxide semiconductors $TiO_2$ and ZnO were used. P25 Degussa® $TiO_2$ and a commercial powder of ZnO (size < 100 nm) were purchased from Sigma Aldrich (Sigma Aldrich, Saint-Quentin Fallavier, France).

#### 4.2.2. Synthesis of the Au/ZnO Photocatalyst

The gold precursor ($HAuCl_4 \cdot 3H_2O$, Sigma Aldrich, Saint-Quentin Fallavier, France) was used without further purification. To synthesize the decorated Au/ZnO material, the photo-deposition method was employed with the $HAuCl_4 \cdot 3H_2O$ precursor in water under UV-A irradiation using a 100W Xenon lamp ($\lambda$ = 365 nm, 17 W/m$^2$) (UVP, Upland, USA) for one hour, as already described [54,68,69]. The amount of Au decoration was fixed to 1% wt to be comparable to the literature data [70]. First, the bare commercial ZnO powder was dispersed in ultra-pure water (1 mg/mL) through sonication for 20 min, without any additional stabilizing or sacrificial agent. Immediately prior to UV-A exposure, the $HAuCl_4 \cdot 3H_2O$ solution (1 mg/mL, 4 mL for 200 mg ZnO) was added to this ZnO solution. After an hour of UV-A irradiation, the resulting blue solution was centrifuged (5000 rpm), washed with distilled water, and dried under vacuum for one hour at room temperature in the absence of light, yielding a blue powder.

### 4.3. Physiochemical Characteristics of Photocatalytic Materials

#### 4.3.1. X-ray Diffraction Analysis (XRD)

The crystalline phase of each photocatalyst was identified by XRD analysis (Panalytical MPDPro, Plaseau, France) at room temperature using Cu K-$\alpha$radiation ($\lambda$ = 1.54 Å). The data were collected over the 2 theta ($\theta$) angular range of 20°–70° (with a step of 0.016 θ/s). The X'pert HighScore software (Malvern Panalytical, https://www.malvernpanalytical.com accessed on 19 July 2023) was used to identify the compounds and for Rietveld refinement. The size of the different crystallites used in the study was evaluated by the Scherrer

equation, applied on the peak with the highest intensity [71]. It should be noted that the obtained values are relative since no integration of the role of the instrument on the peaks widening or internal strain of grains have been considered for these determinations [72].

### 4.3.2. Specific Surface Area Determination

N2 physisorption measurements were made with a Micromeritics ASAP 2020 analyzer (Microméritics, Mérignac, France) at liquid nitrogen temperature. Powders were degassed at 200 °C under high vacuum for 12 h before analyses. A minimum quantity of 200 mg of photocatalyst powder was used to carry out the measurement. The specific area of the three photocatalysts was evaluated using the Brunauer–Emmett–Teller (BET) method in the $P/P° = 0.07–0.2$ range.

### 4.3.3. High-Resolution Transmission Electron Microscopy (HRTEM) Analysis

A JEOL JEM-ARM200F (JEOL LTD, Tokyo, Japan) High-Resolution Transmission Electron Microscope (HRTEM) operating at 200 kV with 1.9 Å resolution was used for TEM and atomic resolution HRTEM imaging.

### 4.3.4. UV-vis Diffuse Reflectance Spectroscopy (DRS) Measurements

DRS allowed the UV-visible spectra of the metal oxide semiconductor-based materials to be obtained. The Perkin Elmer Lambda 950 spectrophotometer (PerkinElmer, Waltham, MA, USA) equipped with snap-in integrating spheres calibrated with a Spectralon® White Diffuse Reflectance Standard was used. The band gap energy (Eg) was estimated based on the absorption spectra and the Kubelka–Munk function [73].

### 4.3.5. Electron Probe Microanalysis (EPMA)

The composition of the Au/ZnO was analyzed by Cameca Electron Probe Microanalysis (EPMA) with a Cameca SXFiveFE (Cameca, Genevillier, France) apparatus (field emission gun 1 to 30 kV, analysis beam diameter down to 70 nm). The system allowed a detection limit of 0.01% for all the elements of the periodic table from Be to U.

### 4.4. Evaluation of Bactericidal and Fungicidal Activity of Photocatalysts

The bactericidal and fungicidal activities of molecules were determined as follows: $TiO_2$ P25 nanoparticles (Degussa, Frankfurt, Germany), ZnO nanoparticles or Au/ZnO nanoparticles were suspended in 50 mL of sterile distilled water and 0.5 mL of the microbial cell suspensions (*E. coli* cells or *A. niger* spores) were added (final concentrations: $10^6$ cells/mL and $10^5$ cells/mL, respectively). The antimicrobial activity of photocatalysts was tested at two concentrations, 10 g/L and 1 g/L. The mixture was gently stirred throughout the incubation period. During each experiment, the test tubes and a control tube without photocatalyst nanoparticles were placed in a sterile flow hood and covered with a Pyrex Lid. Under lighting, test and controls were illuminated with a 8 W black bulb at a light intensity of 5 W/m². In the dark, test and controls tubes were completely covered with cardboard to achieve an intensity of 0 W/m². The intensity of light was measured using a UV-A radiometer Gigahertz-Optik (Gesellschaft mit beschränkter Haftung (GmbH), Türkenfeld, Germany), as described by Verdier et al. [66].

During each experiment, 1 mL was taken from each tube at $t_0 = 0$ min (validation of experimental conditions) and every 30 min for 2 or 4 h. It was diluted ten-fold in sterile distilled water for CFU numerations in the corresponding agar. Petri dishes from *E. coli* and *A. niger* tests were incubated 48 h at $36 \pm 1$ °C and $30 \pm 1$ °C, respectively. The antimicrobial activity (log reduction) was calculated using Equation (1).

$$R = \mathrm{Log}(Ccont) - \mathrm{Log}(Ctest) = \mathrm{Log}\left(\frac{Ccont}{Ctest}\right) \tag{1}$$

where R: $Log_{10}$ Reduction of $TiO_2$, ZnO or Au/ZnO (referred as bactericidal or fungicidal activity in the text), $C_{cont}$: Average concentration of suspension in control tube without

product, expressed in CFU/mL, $C_{test}$: Average concentration of suspension in test tube with $TiO_2$, ZnO or Au/ZnO, expressed in CFU/mL.

Each experiment was repeated twice to ensure reliability. Results are expressed as the mean ± standard deviations of the two independent experiments.

## 5. Conclusions

The main objective of this study was to investigate (i): the difference between the bactericidal effect from one side and fungicidal effect on other side of different photocatalytic products nanoparticles and (ii) to compare the efficacies of different used photocatalysts through stirring experiments allowing a direct contact between the product and cells at low light intensity close to real-world conditions. The obtained results confirmed previous findings on the bactericidal activities of $TiO_2$, ZnO and AuZnO on *E. coli* and highlighted the variation of their efficiencies as a function of concentration, duration of contact, light/dark conditions and especially as a function of microorganisms (spores of *A. niger* vs. *E. coli* in current study). ZnO- and Au-doped ZnO reported stronger bactericidal and fungicidal activities than $TiO_2$ both in the dark and under light conditions. When photocatalyzed, ZnO nanoparticles implanted with gold Au metals reported stronger biocidal activity than undoped ZnO nanoparticles. These findings support the use of Au doped ZnO in paints, coatings and other applications as they present higher bactericidal activities and fair fungicidal activities at light intensities close to those existing in the indoor environment.

**Author Contributions:** M.A.H.: Conceptualization, Methodology, Investigation, Validation, Writing—Original Draft, Visualization; T.V.: Conceptualization, Methodology, Resources, Writing—Review and Editing, Supervision, Project administration; A.B.: Conceptualization, Methodology, Resources, Writing Review and Editing, Supervision; K.C.L.: Methodology, Investigation, Writing—Part of Original draft; O.E.A.: Methodology, Investigation, Writing—Part of Original Draft; K.F.: Writing—Review and Editing; P.F.: Writing—Review and Editing; J.H.: Writing—Review and Editing; C.R.: Writing—Review and Editing; J.-D.B.: Conceptualization, Methodology, Writing—Review and Editing. All authors have read and agreed to the published version of the manuscript.

**Funding:** The authors would like to thank Université Paul Sabatier Toulouse III and the Région Occitanie (Project SoBio2QAI) for their financial support.

**Data Availability Statement:** Data presented in this study are available on request to the corresponding author.

**Conflicts of Interest:** The authors declare no conflict of interest.

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
