# Peer review of "Comparison of Photocatalytic Biocidal Activity of TiO2, ZnO and Au/ZnO on Escherichia coli and on Aspergillus niger under Light Intensity Close to Real-Life Conditions"

_catalysts, doi:10.3390/catal13071139_

Round 1

Reviewer 1 Report

Upon reviewing your manuscript intitled: Comparison of photocatalytic biocidal activity of TiO2, ZnO and Au/ZnO on Escherichia coli and on Aspergillus niger under light intensity close to real-life conditions. I find your work interesting, but I do not believe it can be published in its current form. Therefore, some revisions must be done so that it may be published in this journal. I have the following comments and recommendations.

-The summary is qualitative and does not show the main results. It has initial sentences that would be better in the introduction.

-- The authors do not justify the Au cations concentrations used. Due to the importance for this system, a reasoned explanation must be attached as part of the motivation and justification of the work.

- A detailed explanation of the importance of this work should be provided as part of the motivation. Since several studies have shown results, even better than those discussed here.

- The authors ensure that the refinement of the XRD patterns was carried out using the Rietveld method. In this regard, the authors must include the theoretical standards and the difference between the experimental and theoretical, in addition to including the Bragg positions and Millers indices.

- If refinement of the XRD patterns was done, why was the crystallite size determined using the Scherrer equation?

- The crystallite sizes for the various samples were calculated using the Debye’s Scherrer formula. However, it is well known that the peak broadening is impacted by other factors such as instrument-related broadening, residual stresses in crystals, etc. How did the authors determine or subtract these well-known parameters?

- What does it mean to get a BET specific surface area of 50 m²/g for TiO 2 and ca 14 m²/g for ZnO and Au/ZnO. I suggest a more serious discussion of the characterizations presented. The authors are presenting a comparative study, but throughout the text, the comparisons are not perceived. This should be reviewed carefully to value the research presented.

-From the TEM images, I ask, are the silver particles segregated on the surface of the ZnO particles?

-Error bars must be included in figure 4, and all figures must be improved, from a quality point of view

Reviewer 2 Report

Point 1: Prepare a comparison table for the photocatalytic performance of the current synthesized materials with other photocatalysts.

Point 2: Please indicate the lattice planes corresponding to the lattice spacing in the HRTEM pattern.

Point 3: Why did the authors choose TiO2 as a comparison sample and not another semiconductor with antibacterial activity?

Point 4: Introduction, line 40: “…ions release (Zn2+, Cu2+, Cu+, Ag+), and effective generation of reactive oxygen species due to a higher charge separation .” Authors should be aware of the format in which metal ions are written.

Point 5: Characterization of the photocatalysts, line 102: “Error! Reference source not found. shows the X-Ray diffraction analysis (XRD) patterns of TiO2, ZnO and Au/ZnO metal oxide semiconductors.“ Author's ideographs are unclear. The same problems appear in the latter portion of the text and the author is advised to correct them.

Point 6: Stability and safety are important factors in measuring photocatalysts. The authors are advised to further analyse the stability and metal ion precipitation of photocatalysts.

Point 7: The antibacterial mechanism of ZnO can be divided into two types: Zn2+ active antibacterial mechanism and photocatalytic antibacterial mechanism. The photocatalytic antibacterial mechanism relies mainly on the photocatalytic properties of ZnO and its ability to generate ROS. It is suggested that the authors further demonstrate its photocatalytic mechanism by ESR.

It can be improved.

Reviewer 3 Report

The manuscript describes the antimicrobial activity of TiO2, 24 ZnO and Au/ZnO measured as log reductions of microorganisms E.Coli and Aspergillus niger.

The introduction emphasizes the importance of indoor air quality. However, the experiments were carried out in aqueous suspensions of TiO2, ZnO, and Au/ZnO. This experimental design is far from the real conditions of building materials.

In the bacterial inactivation tests, the maximum exposure time was set to 240 min. Perhaps a longer exposure should be tested to simulate real outdoor conditions in a mild climate. The biocidal effect would probably be increased.  

The main question addressed by the research was the photocatalytic biocidal efficiency of Au-doped ZnO.
In my opinion, the topic is original. It seems that so far the Au-doped ZnO photocatalyst has not been tested for fungicidal activity.
Comparable data on the biocidal efficacy of the new photocatalytic material were obtained
The methodology used is appropriate for the introductory tests. Future studies should include biocidal tests on solid photocatalytic surfaces.

The conclusions are mostly consistent with the obtained results. The following sentence should be improved: “The inclusion of particles such as fungal spores may affect nega-419 tively the efficiency of TiO2 on vegetative cells suggesting that the application of TiO2 nanoparticles in paint [17] is perhaps not the best way to use it.” The references are mostly appropriate. Fig. 5 can be improved by selecting a more appropriate scale on the vertical axis.

Round 2

Reviewer 1 Report

This version of the manuscript can be accepted for publication 

Reviewer 2 Report

The authors have carried out an additional study, corrected the manuscript according to comments. 

Minor editing of English language required.